# Modalities and preferred routes of geographic spread of cholera from endemic areas in eastern Democratic Republic of the Congo

**Harry César Ntumba Kayembe**[1,2]*, **Didier Bompangue**[1,3], **Catherine Linard**[4], **Jérémie Muwonga**[5], **Michel Moutschen**[6], **Hippolyte Situakibanza**[7,8], **Pierre Ozer**[2]

**1** Département des Sciences de Base, Faculté de Médecine, Université de Kinshasa, Kinshasa, Democratic Republic of the Congo, **2** Département des Sciences et gestion de l'environnement, UR SPHERES, Faculté des Sciences, Université de Liège, Arlon, Belgium, **3** Chrono-Environnement, UMR CNRS 6249, Université de Franche-Comté, Besançon, France, **4** Département de Géographie, Université de Namur, Namur, Belgium, **5** Département de Biologie Médicale, Faculté de Médecine, Université de Kinshasa, Kinshasa, Democratic Republic of the Congo, **6** Département des Sciences cliniques, Immunopathologie–Maladies infectieuses et Médecine interne générale, Université de Liège, Liege, Belgium, **7** Département de Médecine interne, Faculté de Médecine, Université de Kinshasa, Kinshasa, Democratic Republic of the Congo, **8** Département de Parasitologie et Médecine tropicale, Faculté de Médecine, Université de Kinshasa, Kinshasa, Democratic Republic of the Congo

* harry.kayembe@unikin.ac.cd

**Data Availability Statement:** All relevant data are within the paper and its Supporting Information files.

## Abstract

Cholera is endemic along the Great Lakes Region, in eastern Democratic Republic of the Congo (DRC). From these endemic areas, also under perpetual conflicts, outbreaks spread to other areas. However, the main routes of propagation remain unclear. This research aimed to explore the modalities and likely main routes of geographic spread of cholera from endemic areas in eastern DRC. We used historical reconstruction of major outbreak expansions of cholera since its introduction in eastern DRC, maps of distribution and spatiotemporal cluster detection analyses of cholera data from passive surveillance (2000–2017) to describe the spread dynamics of cholera from eastern DRC. Four modalities of geographic spread and their likely main routes from the source areas of epidemics to other areas were identified: in endemic eastern provinces, and in non-endemic provinces of eastern, central and western DRC. Using non-parametric statistics, we found that the higher the number of conflict events reported in eastern DRC, the greater the geographic spread of cholera across the country. The present study revealed that the dynamics of the spread of cholera follow a fairly well-defined spatial logic and can therefore be predicted.

## Introduction

Cholera is an epidemic-prone acute diarrheal disease caused by a well-recognized pathogen for humans "*Vibrio cholerae*" [1]. It is transmitted through ingestion of water or food

**Funding:** The author(s) received no specific funding for this work.

**Competing interests:** The authors have declared that no competing interests exist.

contaminated with toxigenic forms of the bacterium [2]. Human to human contamination is also reported following limited access to clean drinking water and poor sanitation [2, 3].

The disease remains a worldwide major public health threat, especially in Sub-Saharan countries [4]. Large epidemics are annually recorded, particularly in the Democratic Republic of the Congo (DRC). According to World Health Organization (WHO), the DRC reported more than 420,000 suspected cholera cases during the 2000–2017 period. It accounted for nearly 10% and 17% of the global and African cholera-related morbidity, respectively [4].

Cholera was first reported in eastern DRC in 1978. Cases were determined to be imported from Kigoma in Tanzania to Kalemie, both bordering Lake Tanganyika, then the disease spread along the Great Lakes Region (GLR) [5, 6]. Over two decades later, the cholera control model in the DRC was essentially based on one-off response interventions in areas affected [5–7], and less attention has been paid to understanding the dynamics of the disease. Therefore, the role of certain factors, especially during inter-epidemic periods, in the recurrence of cholera outbreaks has not yet been investigated. Thus, the occurrence of epidemics seemed unpredictable, anarchic, even without spatio-temporal logic.

Since the 2000s, several epidemiological and ecological studies identified the GLR in eastern DRC as cholera endemic with stable transmission foci bordering lakes [8–10]. The role of seasonal population movements driven by fishing and commercial activities has been suggested in the spread of cholera outbreaks from hotspots to other areas [10, 11]. Furthermore, cholera outbreaks are infrequently reported in other parts of the country, particularly in western DRC. The latter is affected by the spatial spread of cholera from the GLR to the major cities upstream of the Congo River, before contiguously reaching the areas of the downstream provinces, and then the capital of Kinshasa and the mouth of the Congo River [12, 13]. During the last decade, this propagation pattern was observed in 2011–2012 and 2015–2017. A few phylogenetic analyses have recently confirmed the westward spread dynamics of cholera from the GLR [14, 15].

Despite the growing body of evidence on the spread of cholera in the DRC, the propagation routes, particularly those reaching areas outside the endemic eastern provinces, and their original epidemic foci remain unclear. We therefore hypothesized that there are several modalities and main routes of geographic spread of cholera from endemic lakeside areas.

The eastern DRC is also known as an active conflict zone for decades. War and conflict lead to massive population displacement, the collapse of health systems, and the breakdown of surveillance, early warning and response systems [16]. Conflict-affected populations are at risk of exposure to inadequate access to water, sanitation, and hygiene (WASH) facilities and resources [17, 18], which further increases the risk of epidemics [19]. While the impact of conflicts on the geographic spread of cholera was demonstrated elsewhere [20–22], epidemic reactivations of cholera in health zones (HZs) were suggested to be favored by conflicts in eastern DRC [9]. Thus, in our context, conflict events would amplify the spatial spread of cholera outbreaks out of endemic eastern provinces.

In this study, we aimed to explore the modalities and likely main routes of geographic spread of cholera from endemic areas in eastern DRC.

## Data and methods

### Study setting

The DRC is located in Central Africa. It has a total area of 2,345,000 km$^2$ and a population of 86,895,208 inhabitants [23] with a population density estimated at 37.06 people/km$^2$. Overall, the proportion of household population using an improved source of drinking water and improved sanitation facilities represents 59% and 32%, respectively [24].

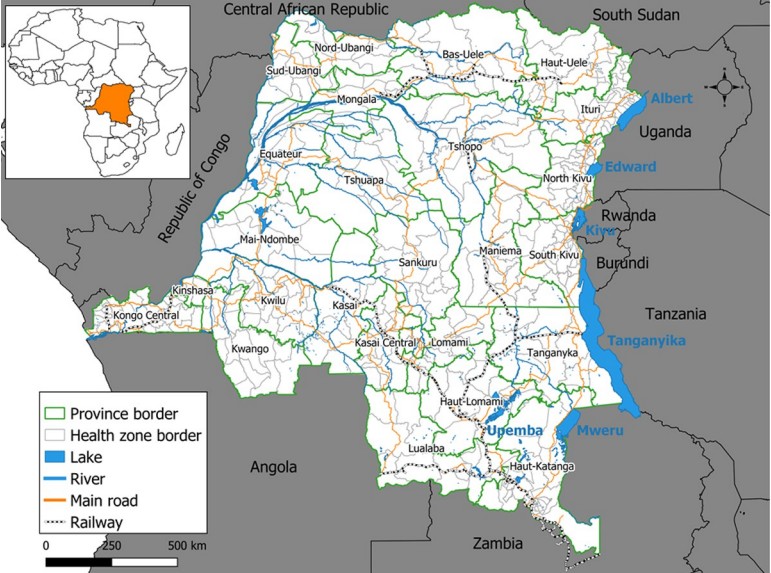

**Fig 1. Administrative map of the DRC.**

The country is currently subdivided into 26 administrative provinces also corresponding to provincial health divisions and 518 HZs (Fig 1). The HZ is the operational level of the health pyramid, while the provincial health division (intermediate level) play a technical and logistical role and the Minister of Health's office and the general secretariat with 13 directorates and 52 specialized programs (central level) sets standards. Each HZ counts an average population of 100,000 to 300,000 inhabitants and consists of one reference general hospital and 15–20 health centers.

## Data collection

**Information on major outbreak expansions of cholera documented since its introduction in eastern DRC.** historical reconstruction was conducted to summarize major outbreak expansions of cholera since its introduction in eastern DRC in 1978 using queries on PubMed, Google and Google scholar. We used the following key words: "Cholera OR Vibrio cholerae" AND "Democratic Republic of the Congo". Peer-reviewed and non-peer-reviewed articles published in English or French and reports from humanitarian agencies focused on the period 1970–2017 were searched. Only full texts that addressed the diffusion processes were considered.

**Epidemiological surveillance data.** Cholera is of 15 diseases with epidemic potential reported weekly by the National Integrated Disease Surveillance and Response system (IDSRS), established in 2000 through the partnership between the DRC Ministry of Health and WHO. Suspected cholera cases and deaths are identified through a syndromic approach. According to WHO, a suspected cholera case is defined as follows: "In areas where a cholera outbreak has not been declared: Any patient aged 2 years and older presenting with acute watery diarrhea and severe dehydration or dying from acute watery diarrhea. In areas where a cholera outbreak is declared: any person presenting with or dying from acute watery diarrhea" [25]. Each new outbreak is confirmed by culture and isolation of *Vibrio cholerae* O1 from stool samples [26].

Cholera morbidity and mortality data are recorded by medical officers in each cholera treatment center (CTC) using line lists [25] on hard format, and then aggregated electronically by

Ministry of Health officials at HZ scale. These data are transmitted weekly to the provincial health divisions, before being centralized at the General Direction of Disease Control (https://dhis2.fbp-rdc.org/). The latter is the organ of the Ministry of Health that manages the IDSRS.

Here, we used weekly suspected cholera cases data collected at the HZ level from January 2000 to December 2017. Also, we defined cholera outbreaks as periods with the doubling of suspected cases in endemic HZs or the occurrence of at least one case in non-endemic HZs over three consecutive weeks (the end being respectively this 3-week period with the lowest threshold or no case reports).

**Conflict data.**   Data on conflict events were derived from a leading publicly available conflict events dataset which provide high-resolution georeferenced and disaggregated data, the Armed Conflict Location and Event Dataset (ACLED). The ACLED reports both violent and non-violent events, without any restrictive fatalities threshold, at a daily step from cross-checking of multiple information sources [27]. This concerns multiple geographic scale (local, regional, national and continental) media, reports from non-governmental or international organizations in addition to media reporting, selected social media accounts (Twitter and Telegram) and partnerships with local conflict observatories in hard-to-access cases [27, 28]. Types of conflict events collected are: battles, explosions/remote violence, violence against civilians, protests, riots, and strategic developments [27]. In this study, we focused on conflict events reported during 2000–2017. They were aggregated at a weekly time step and the HZ scale.

**GIS data.**   Free open shapefiles of large-scale boundaries of African countries [29] and the DRC at the HZ level [30] were obtained from the open access data platform "The Humanitarian Data Exchange".

## Data analysis

Major outbreak expansions of cholera documented since its introduction in eastern DRC were summarized by cross-checking of information from published articles, epidemiological reports, and historical records of cholera outbreaks. We then mapped the propagation routes. Other maps of distribution of cholera cases per week and per HZ were generated to describe the geographic spread modalities of cholera from endemic areas in eastern DRC. All maps were produced using Quantum GIS version 3.8.3.

To explore the likely main routes of geographic spread of cholera outbreaks from endemic areas, the Kulldorff's retrospective space-time permutation scan statistic was implemented to detect spatiotemporal clusters of annual cholera cases using SaTScan software version 9.6 [31]. This model does not require population-at-risk data, only the number of cases to estimate the expected number of cases. The scanning window is defined using a large number of overlapping cylinders which the circular base and the height represent respectively geographical location and time. Adjustments are made using a large number of random permutations of the spatial and temporal attributes of each observation in the dataset being studied. The most likely cluster is estimated for each permutation of the simulated dataset. Statistical significance is assessed using the Monte Carlo hypothesis testing [32]. In our study, the *p-value* was estimated using 999 Monte Carlo simulations. As Horwood et al. [33], after identifying very large spatiotemporal clusters containing a number of statistically significant sub-clusters in preliminary analyses, we set the maximum spatial window as a circle with a 125 km radius.

In addition, as eastern DRC is heavily affected by armed conflicts, we compared differences in the weekly number of conflict events reported in cholera-endemic provinces (Haut Katanga, Haut Lomami, Ituri, North Kivu, South Kivu, and Tanganyika) according to the observed geographic spread modalities of outbreaks. Given this variable followed a skewed distribution, we

used medians to assess whether differences found in our study were statistically significant using the Kruskal-Wallis test and Wilcoxon-rank sum tests. The statistical analyses were performed with R$^®$ version 3.6.1.

### Ethics approval

Ethics approval was not required because this study was carried out with routinely collected surveillance data and aggregated at the HZ level.

## Results

### Summary of major outbreak expansions of cholera since its introduction in eastern DRC

A list of 1,288 studies and reports were identified using the search terms. We removed 384 records due to duplication, and then we screened 904 records of which 889 were considered not relevant according to titles and abstracts or full texts that did not address the dynamics of the spread of cholera in and from eastern DRC. 15 studies and reports were eligible and detailed in Table 1.

The introduction of cholera in 1978 in eastern DRC resulted from the importation of the disease into East Africa via Ethiopia from the Middle East in 1970 (number 1, Fig 2A) [14, 34–36]. Since then, neighboring countries were affected: Djibouti, Somalia, Sudan, Uganda and Kenya (number 2, Fig 2A) [34, 35]. In 1974, cholera spread from the shores of Lake Victoria to the Indian Ocean coastline and then reached Dar-es-Salaam (1977) (number 3, Fig 2A) [35].

Fig 2B summarized the following diffusion dynamics: 4 –From the Tanzanian cost to inland areas, Kigoma (April, 1978) [34, 35]; 5 –Northward from Kigoma to the Congolese lakeside areas of lakes Tanganyika, Kivu, Edward and Albert between June and December [5, 34, 35]; 6 –Southward from Kigoma to Kalemie (May, 1978), and then to the shores of Lake Mweru (1981) [5, 6, 34, 35]; 7 –Westward from the GLR to Kisangani (March, 1979) [6, 35]; 8 –Outbreaks occurred in temporary refugee camps after fleeing from the former camps in eastern DRC and other provinces (Equateur and Kinshasa) (1997) [37–39]; 9 –From Bukama to Mbuji Mayi and Lubumbashi (2002) [8]; 10 and 10'–Westward to non-endemic provinces of western DRC (2011–2012 and 2015–2017) [12, 13, 15]. Neighboring countries of the West Congo Basin were affected during these outbreak expansions [14, 15, 40, 41].

### Geographic spread of cholera from endemic eastern areas: Modalities and likely main routes

Using maps of propagation of cholera cases per week and per HZ (S1 Fig), the geographic spread modalities of the disease were described as follows: (i) The spread within endemic eastern provinces including: along lakeside HZs in the same province or in two bordering provinces, from lake areas to adjacent non-endemic HZs (directly sharing their borders) in the same and/or neighboring province, and to non-adjacent HZs (not directly sharing their borders) in the same and/or neighboring province; (ii) The spread outside the endemic provinces to HZs in non-endemic provinces of eastern DRC; (iii) The spread outside the endemic provinces to the HZs in non-endemic provinces of central DRC; (iv) The spread outside the endemic provinces to the HZs in non-endemic provinces of western DRC.

Fig 3 described the spatiotemporal clusters of cholera cases identified and the likely routes of spread of outbreaks from endemic eastern areas during 2000–2005. Details on space-time clustering results were summarized in S1–S6 Tables. Cholera most likely spread to non-endemic eastern provinces (Maniema and Tshopo) from areas around Lake Kivu through the

**Table 1. Summaries of the 15 studies and reports considered as relevant in our study.**

| Citation | Type | Periods | Geographical context | Approach |
|---|---|---|---|---|
| Schyns.1979. Cholera in Eastern Zaire, 1978 | Research | 1978 | Eastern Zaïre | Reconstruction of the spatio-temporal evolution and epidemiological analysis |
| Malengreau. 1979. The cholera epidemic in Eastern Zaire in 1978. | Research | 1978 | Eastern Zaïre | Reconstruction of the spatio-temporal evolution and epidemiological analysis |
| Carme. 1983. L'implantation et l'extension du choléra en Afrique Noire: 1970–1980 | Research | 1970–1980 | West Africa, East Africa, and Southern Africa | Reconstruction of the spatio-temporal evolution |
| Rémy & Dejours. 1988. L'Africanisation du choléra | Research | 1970–1985 | North East Africa, East Central Africa, Southern Africa, West Central Africa, and West Africa | Reconstruction of the spatio-temporal evolution |
| Swerdlow & Isaäcson. 1994. The Epidemiology of Cholera in Africa. | Research | 1972–1991 | Africa | Reconstruction of the epidemic and epidemiological analyses |
| World Health Organization. 1997—Health situation in Rwandan refugee camp in Zaire. | Report of disease outbreak | 1997 | North-eastern Zaïre | Reconstruction and description of the epidemic |
| Centers for Disease Control and Prevention. 1998. Cholera Outbreak among Rwandan Refugees—Democratic Republic of Congo, April 1997. | Report of disease outbreak | 1997 | North-eastern Zaïre | Description of the epidemic |
| World Health Organization. 1997—Cholera in Zaire. | Report of disease outbreak | 1997 | North-eastern Zaïre | Description of the epidemic |
| Bompangue. 2008. Lakes as Source of Cholera Outbreaks, Democratic Republic of Congo. | Research | 2002–2005 | South-Eastern and Central DRC | Reconstruction of the spatio-temporal evolution and epidemiological analyses |
| UNICEF. 2011. UNICEF fights "one of the worst ever" cholera outbreaks in West and Central Africa | Report of disease outbreak | 2011 | The West Congo Basin (DRC, Congo and the Central African Republic) | Description of the epidemic |
| Bompangue. 2012. Cholera ante portas–The re-emergence of cholera in Kinshasa after a ten-year hiatus | Research | 1996–2011 | Kinshasa (western DRC) | Reconstruction of the spatio-temporal evolution (2011 epidemic) and epidemiological analyses |
| Weill. 2017. Genomic history of the seventh pandemic of cholera in Africa | Research | 1966–2014 | Africa | Reconstruction of the spatio-temporal evolution through genomic analysis |
| Moore. 2018. Epidemiological study of cholera hotspots and epidemiological basins in East and Southern Africa. In-depth report on cholera epidemiology in Angola | Report | 2006–2018 | Eastern DRC, western DRC, and Angola | Epidemiological analyses |
| Ingelbeen. 2019. Recurrent cholera outbreaks, Democratic Republic of the Congo, 2008–2017 | Research | 2008–2017 | DRC | Reconstruction of the spatio-temporal evolution and epidemiological analyses |
| Breurec. 2021. Seventh pandemic *Vibrio cholerae* O1 sublineages, Central African Republic | Research | 1997–2016 | Eastern DRC, western DRC, and Central African Republic | Reconstruction of the spatio-temporal evolution through genomic analysis |

westernmost HZs of North Kivu, Pinga and Walikale. Those provinces were also affected in a westerly direction from lakeside HZs of Lake Albert (Ituri) and Lake Tanganyika. The disease reached the city of Lubumbashi in southeastern DRC from the shores of lakes Mweru (Haut Katanga), Tanganyika, and Upemba (Haut Lomami). The latter was at the origin of the spread of cholera to the central part of the DRC. Clusters detected in western DRC in 2000 were related to the dynamics that started in previous years.

Fig 4 depicted the spatiotemporal clusters of cholera cases and the likely routes of spread of outbreaks from endemic eastern areas during 2006–2011. Detailed results of cluster detection analyses were shown in S7–S12 Tables. The western part of Tanganyika province was successively affected from the shores of Lake Tanganyika, Lake Mweru and Lake Upemba, and then the disease progressed to the southern part of Maniema (2006) and the city of Kindu (2008). Clusters detected in southwestern DRC in 2006 were epidemiologically linked to the outbreak in neighboring country, Angola. The spread of cholera to the city of Kisangani (Tshopo) and

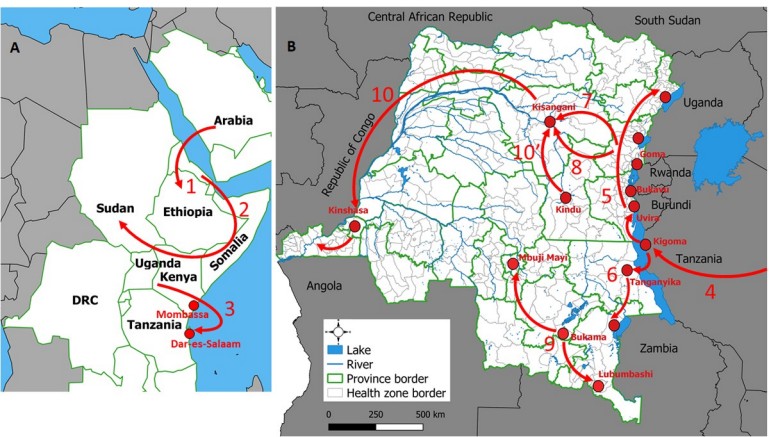

**Fig 2.** A and B. Major outbreaks expansions of cholera documented since its introduction in eastern DRC in 1978. Sources: Fig 2A [29] and Fig 2B [30].

then to non-endemic western provinces was most likely initiated around Lake Kivu via the Walikale HZ in North Kivu (2010–2011).

Fig 5 illustrated the space-time clusters of cholera cases detected and the likely routes of spread of outbreaks from endemic eastern areas during 2012–2017. The clusters were detailed in S13–S18 Tables. All the spatial spread modalities were observed during 2012–2017. Beyond the east-west propagation, several most likely routes of cholera spread to Kisangani were identified: from Kindu (initiated around Lake Kivu and then via the north of Maniema), from lake areas bordering lakes Kivu and Albert respectively via Pinga and Walikale HZs, and the western part of Ituri. The western HZs of Tanganyika province were affected from lakes Upemba and Tanganyika by northward and westward spread, respectively. A southward spread of cholera was repeatedly observed from the lakeside HZs of Upemba to Lubumbashi before reaching other HZs in Lualaba province. Outbreak affecting provinces of Lomami and Kasaï Oriental was also originated around Lake Upemba.

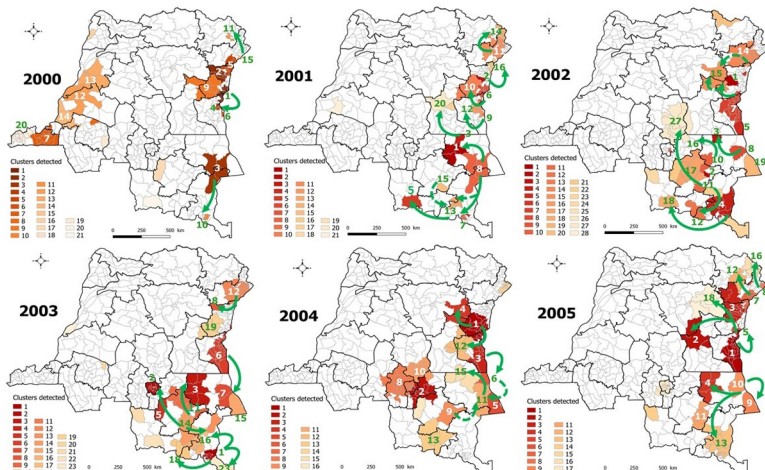

**Fig 3. Spatiotemporal clusters of cholera cases, DRC, 2000–2005.** Arrows indicate the likely routes of spread. Dashed arrows represent two probable routes at the same period. Republished from [30] under a CC BY license, with permission from [Claire Halleux], original copyright [2021].

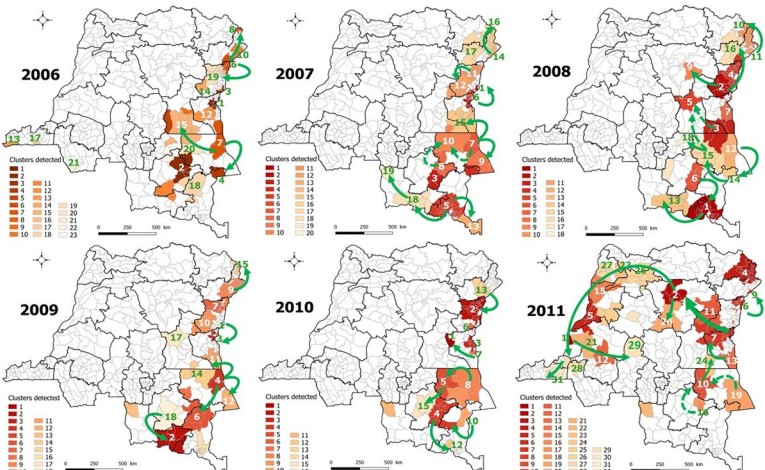

**Fig 4. Spatiotemporal clusters of cholera cases, DRC, 2006–2011.** Arrows indicate the likely routes of spread. Dashed arrows represent two probable routes at the same period. Republished from [30] under a CC BY license, with permission from [Claire Halleux], original copyright [2021].

## Comparison of the weekly number of conflicts reported in endemic eastern provinces according to geographic spread modalities of cholera outbreaks

The Kruskal-Wallis test showed that the median weekly number of conflict events reported in endemic eastern provinces was significantly greater for Modality IV than for Modality II and Modality I in North Kivu ($X^2$ = 159.58, df = 2, $p < 0.001$) and South Kivu ($X^2$ = 100.97, df = 2, $p < 0.001$) (Fig 6). Moreover, the Wilcoxon-rank sum tests for each pairwise comparison of geographic spread modalities of cholera confirmed that all differences between the weekly number of conflict events reported in those provinces were statistically significant (Table 2).

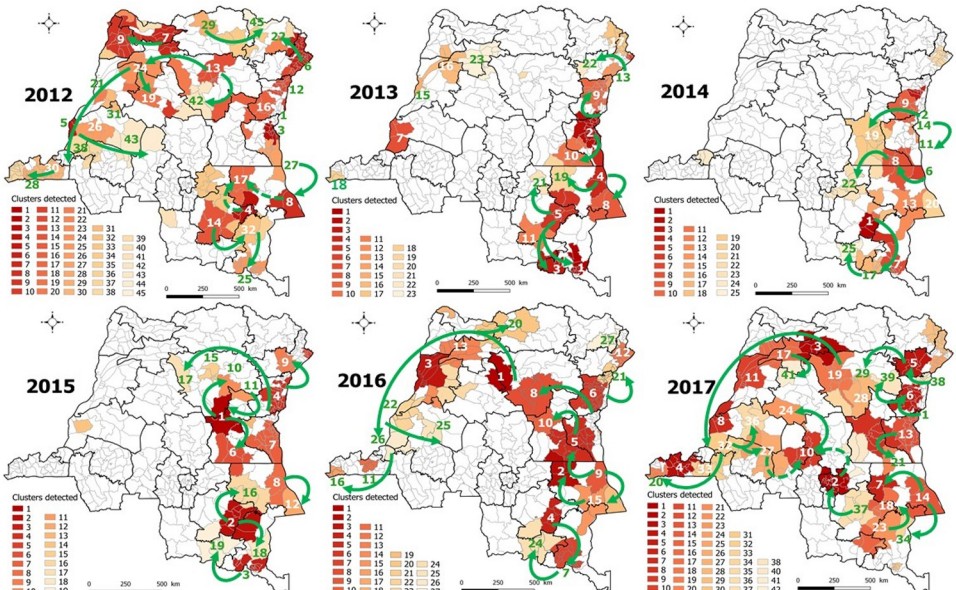

**Fig 5. Spatiotemporal clusters of cholera cases, DRC, 2012–2017.** Arrows indicate the likely routes of spread. Dashed arrows represent two probable routes at the same period. Republished from [30] under a CC BY license, with permission from [Claire Halleux], original copyright [2021].

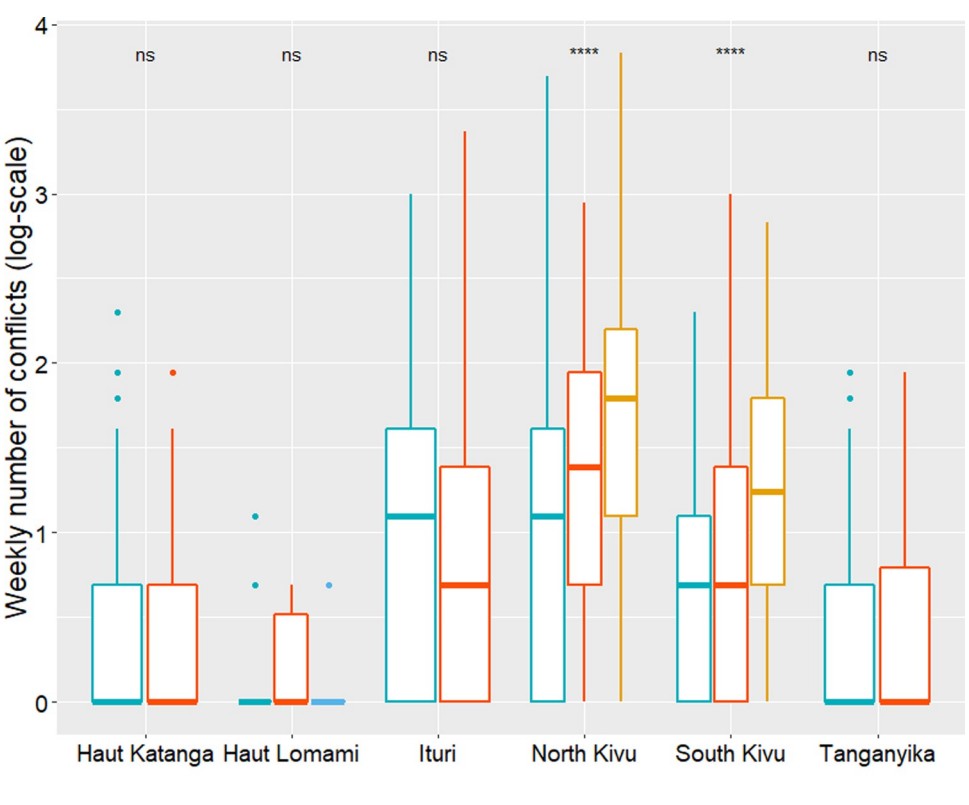

**Fig 6. Boxplots of the weekly numbers of conflict events reported in eastern endemic provinces by geographic spread modalities of cholera outbreaks.**

## Discussion

The present study described four geographic spread modalities of cholera outbreaks from endemic eastern foci and their main likely propagation routes according to the affected regions at the national level: around lake areas and to their surroundings in endemic eastern provinces, to other areas respectively located into eastern, central and western non-endemic provinces. In addition, according to the observed spread modalities, the higher the weekly number of conflict events reported in endemic eastern provinces (North and South Kivu), the greater the geographic spread of cholera across the country.

**Table 2. Wilcoxon-rank sum test results.**

| Pairwise comparisons | Medians | Test statistics (W) | p-value |
|---|---|---|---|
| *North Kivu* | | | |
| Modality I–Modality II | 0.00–0.69 | 21411 | < 0.001 |
| Modality I–Modality IV | 0.00–1.79 | 13510 | < 0.001 |
| Modality II–Modality IV | 0.69–1.79 | 15630 | < 0.001 |
| *South Kivu* | | | |
| Modality I–Modality II | 0.00–0.00 | 22695 | 0.004 |
| Modality I–Modality IV | 0.00–1.10 | 5474.5 | < 0.001 |
| Modality II–Modality IV | 0.00–1.10 | 9165 | < 0.001 |

Understanding the spread dynamics of infectious diseases is of utmost importance in the anticipation and control of their epidemics [42]. In this study, although the processes of diffusion of cholera epidemics from source areas are multidirectional, main likely routes of spread have emerged as preferential trajectories. The propagation route from the areas around Lake Kivu to the westernmost HZs of North Kivu, Pinga and Walikale, via the main roads would probably be the most involved in the spread of cholera out of endemic eastern DRC to the northern part of Maniema, including the city of Kindu, and to the city of Kisangani (Tshopo). The latter could also be affected by the westward spread of the disease from the shores of Lake Albert in Ituri through the main road network. Furthermore, the lakeside areas of Haut Lomami and Tanganyika would be the most probable original foci of outbreaks affecting the western part of Tanganyika province (Ankoro, Manono, Kabalo, Mbulala, and Kongolo). From there, the disease could gradually progress to the southern part of Maniema, then to Kindu as well as to Kisangani. Note that Kisangani serves as a transport hub in the east-west spread of cholera reported in the literature [12–15].

Other main likely routes of disease spread were observed. Lubumbashi, the capital of Haut Katanga in southeastern DRC, would be affected by the southward spread of outbreaks from areas bordering either Lake Upemba or Lake Mweru, most likely via major roads. The lake areas of Haut Lomami would also be the main source of epidemics affecting the eastern part of Grand Kasai in central DRC through the railways.

The main propagation routes of cholera described as preferential trajectories through major roads, rivers, and railways are consistent with other findings which considered these transport networks as key drivers of the disease spread through population movements [8, 13, 43, 44]. Targeted prevention and control efforts should also take into account these preferential trajectories from known endemic areas in order to effectively eliminate the disease in the DRC.

The geographic spread of cholera epidemics out of endemic eastern DRC may be linked to the exacerbation of conflict events reported in this region, particularly in North and South Kivu. Interestingly, this finding is supported by evidence of increased spatial spread of infectious diseases in countries heavily affected by conflicts [20–22]. The humanitarian crisis in Yemen, resulting from the ongoing devastating war, led to the largest and fastest spreading cholera epidemic worldwide [20, 45–47]. In our context, these dynamics of spread would be consecutive to the migration flows of thousands of internally displaced persons (IDPs) due to war and conflict situations. The latter also negatively impact the health system and other infrastructures that lose the capacity to provide even basic services [16]. As a result, conflict-affected populations are at increased risk of food insecurity, shortages of potable water, and poor sanitation and hygiene practices in host sites due to rapid and massive relocation [17, 18]. It has been shown previously that a one-day interruption in water supply is followed by a substantial increase in the incidence rate of suspected cholera cases within 12 days [48].

Nevertheless, we also found other propagation routes of epidemics out of endemic eastern provinces initiated from areas less affected by conflicts. As previously suggested, these dynamics of spread are likely associated with seasonal fishing and fish trading activities [10, 11]. Although this explained the spread of cholera to the eastern part of the central provinces of the DRC in 2002 [8, 49], an explosive outbreak occurred 15 years later in this region, following a similar pattern, under a conflict-fueled humanitarian crisis that led to the collapse of almost the entire health system [50]. These observations corroborate the idea that the risk of geographic spread of cholera is multifactorial. In addition to conflict events [16, 51], it may require the interaction of ecological, socioeconomic and behavioral factors involving human migratory dynamics [52].

This study presented a number of limitations. First, we used data on suspected cholera cases. It may have over or underestimated the true burden of cholera. However, a recent assessment on the level of adequacy of the 15 weekly reported epidemic prone diseases monitored by the DRC's surveillance system demonstrated that the use of these data may be relevant for epidemiological or public health research purposes [53]. Furthermore, another assessment of IDSR key performance indicators showed that the DRC figures among the African countries with high coverage of IDSR implementation at subnational [54]. Second, the lack of genomic data from *Vibrio cholerae* O1 isolates. Phylogenetic analysis would have allowed determination of stable propagation routes of circulating cholera strains from the different endemic foci. Nonetheless, to our knowledge, this is the first epidemiological study carried out in such a long period (18 years) that explored the geographic spread modalities and their most likely preferential trajectories from endemic eastern areas. This provides additional understanding elements to the current state of knowledge on the spread dynamics of cholera in the DRC. Third, we just looked at the weekly number of conflict events in endemic provinces, without including non-endemic conflict events. Considering that endemic provinces account for 73% of reported conflict events in the country [27], two-thirds of which occur in North and South Kivu, we hypothesized that the exacerbation of conflicts in eastern DRC would be responsible for the spread of cholera to far-off areas in non-endemic provinces. Our results revealed that the most likely routes of spread involved in these dynamics would originate mainly from endemic areas in North and South Kivu. However, the few dynamics of spread out of endemic provinces initiated from eastern areas less affected by conflicts further highlight the key role of population movements related to IDPs on the one hand, and commercial activities on the other. There is need for futures studies to explore how conjunctural and structural population movements may affect the geographic spread of cholera from endemic eastern areas.

## Conclusions

The modalities and the likely main routes of geographic spread of cholera outbreaks from the source areas described in this study highlight that the dynamics of the disease's expansion follow a fairly well-defined spatial logic, and can therefore be predicted. These results could contribute to the development of a plan to build resilience in HZs iteratively affected by epidemic waves spreading from endemic areas to achieve the 2030 goals of reducing cholera as a major public health threat [55]. Further phylogenetic researches will help to confirm the likely preferred routes of spread of cholera epidemics identified in the DRC.

The exacerbation of conflict events reported in eastern DRC is most likely associated with the spread of outbreaks affecting areas increasingly distant from endemic foci. This implies the possibility of setting up early warning systems including monitoring of conflict dynamics to anticipate the risk of geographic spread of cholera in the DRC.

## Supporting information

**S1 Fig. Distribution of weekly suspected cholera cases by health zone, DRC, 2000–2017.** Republished from [30] under a CC BY license, with permission from [Claire Halleux], original copyright [2021].
(DOCX)

**S1 Data.**
(TXT)

**S2 Data.**
(TXT)

**S1 Table. Spatiotemporal clusters of cholera cases, DRC, 2000.**
(DOCX)

**S2 Table. Spatiotemporal clusters of cholera cases, DRC, 2001.**
(DOCX)

**S3 Table. Spatiotemporal clusters of cholera cases, DRC, 2002.**
(DOCX)

**S4 Table. Spatiotemporal clusters of cholera cases, DRC, 2003.**
(DOCX)

**S5 Table. Spatiotemporal clusters of cholera cases, DRC, 2004.**
(DOCX)

**S6 Table. Spatiotemporal clusters of cholera cases, DRC, 2005.**
(DOCX)

**S7 Table. Spatiotemporal clusters of cholera cases, DRC, 2006.**
(DOCX)

**S8 Table. Spatiotemporal clusters of cholera cases, DRC, 2007.**
(DOCX)

**S9 Table. Spatiotemporal clusters of cholera cases, DRC, 2008.**
(DOCX)

**S10 Table. Spatiotemporal clusters of cholera cases, DRC, 2009.**
(DOCX)

**S11 Table. Spatiotemporal clusters of cholera cases, DRC, 2010.**
(DOCX)

**S12 Table. Spatiotemporal clusters of cholera cases, DRC, 2011.**
(DOCX)

**S13 Table. Spatiotemporal clusters of cholera cases, DRC, 2012.**
(DOCX)

**S14 Table. Spatiotemporal clusters of cholera cases, DRC, 2013.**
(DOCX)

**S15 Table. Spatiotemporal clusters of cholera cases, DRC, 2014.**
(DOCX)

**S16 Table. Spatiotemporal clusters of cholera cases, DRC, 2015.**
(DOCX)

**S17 Table. Spatiotemporal clusters of cholera cases, DRC, 2016.**
(DOCX)

**S18 Table. Spatiotemporal clusters of cholera cases, DRC, 2017.**
(DOCX)

## Acknowledgments

We are grateful to the Académie de Recherche et d'Enseignement Supérieur (ARES). We also thank Dr Bien-Aimé Mandja for his participation in the analysis and interpretation of the

data, all staff of the Ecology and Control of Infectious Diseases Unit (Service d'Ecologie et Contrôle des Maladies Infectieuses) for their support, and Claire Halleux from OpenStreetMap RDC for providing access to the DRC dataset at the HZ level under the CC BY 4.0 license.

## Author Contributions

**Conceptualization:** Harry César Ntumba Kayembe, Didier Bompangue, Catherine Linard, Jérémie Muwonga, Michel Moutschen, Hippolyte Situakibanza, Pierre Ozer.

**Data curation:** Harry César Ntumba Kayembe.

**Formal analysis:** Harry César Ntumba Kayembe.

**Methodology:** Harry César Ntumba Kayembe, Didier Bompangue, Catherine Linard, Jérémie Muwonga, Hippolyte Situakibanza, Pierre Ozer.

**Software:** Harry César Ntumba Kayembe.

**Validation:** Catherine Linard, Hippolyte Situakibanza, Pierre Ozer.

**Visualization:** Harry César Ntumba Kayembe.

**Writing – original draft:** Harry César Ntumba Kayembe.

**Writing – review & editing:** Catherine Linard, Hippolyte Situakibanza, Pierre Ozer.

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
