## [Decision Letter · Decision Letter 0]

25 Nov 2021

PONE-D-21-31003Modalities and preferred routes of geographic spread of cholera from endemic areas in eastern Democratic Republic of the CongoPLOS ONE

Dear Dr. Kayembe,

Thank you for submitting your manuscript to PLOS ONE. After careful consideration, we feel that it has merit but does not fully meet PLOS ONE’s publication criteria as it currently stands. Therefore, we invite you to submit a revised version of the manuscript that addresses the points raised during the review process.

Please consider the comments of the reviewer to improve the manuscript. Please submit your revised manuscript by Jan 09 2022 11:59PM. If you will need more time than this to complete your revisions, please reply to this message or contact the journal office at plosone@plos.org. Please include the following items when submitting your revised manuscript:A rebuttal letter that responds to each point raised by the academic editor and reviewer(s). You should upload this letter as a separate file labeled 'Response to Reviewers'.A marked-up copy of your manuscript that highlights changes made to the original version. You should upload this as a separate file labeled 'Revised Manuscript with Track Changes'.An unmarked version of your revised paper without tracked changes. You should upload this as a separate file labeled 'Manuscript'.

We look forward to receiving your revised manuscript.

Kind regards,

Axel Cloeckaert

Academic Editor

PLOS ONE

Journal Requirements:

3. We note that Figure(s) 1, 2, 3, 4 and 5 in your submission contain [map/satellite] images which may be copyrighted. All PLOS content is published under the Creative Commons Attribution License (CC BY 4.0), which means that the manuscript, images, and Supporting Information files will be freely available online, and any third party is permitted to access, download, copy, distribute, and use these materials in any way, even commercially, with proper attribution. For these reasons, we cannot publish previously copyrighted maps or satellite images created using proprietary data, such as Google software (Google Maps, Street View, and Earth). For more information, see our copyright guidelines: http://journals.plos.org/plosone/s/licenses-and-copyright.

1. You may seek permission from the original copyright holder of Figure(s) 1, 2, 3, 4 and 5 to publish the content specifically under the CC BY 4.0 license.  

Reviewers' comments:

Reviewer's Responses to Questions

**Comments to the Author**

1. Is the manuscript technically sound, and do the data support the conclusions?

Reviewer #1: Yes

2. Has the statistical analysis been performed appropriately and rigorously? 

Reviewer #1: Yes

3. Have the authors made all data underlying the findings in their manuscript fully available?

Reviewer #1: Yes

4. Is the manuscript presented in an intelligible fashion and written in standard English?

Reviewer #1: Yes

5. Review Comments to the Author

Reviewer #1: Thank you very much for this interesting and important research, it was a pleasure to review and it was great to read your work. As you mention in your introduction, cholera in the DRC is at alarming levels and research on cholera dynamics in the DRC is essential. Congratulations on finding statistically significant transmission routes for the spread of cholera in the DRC.

The work is presented in a clear way and is easy to follow and you make some reasoned conclusions at the end in terms of disease and disaster planning and resilience, based on your findings. I have a few minor comments below.

Comments

Line 58-62: “no particular attention has been paid to understanding the dynamics of the disease”, I am not sure this is justified. In term of cholera, we can definitely do more, but I am not quite sure you could say no attention has been paid to this. There are several excellent papers looking at cholera risk factors and transmission, some of which in the DRC and many you cite in the next paragraph.

Line 78: Would be good to include an extra sentence on why conflicts amplify the spread. E.g., disruption to WASH or healthcare, displacement etc.

Line 93: Not sure about the ’s on health, double check that is grammatically sound.

The figures are great, informative and add to your narrative but they are a little blurry on my screen. Maybe just check the quality.

Line 102-107: Why did you search only cholera and DRC, why not Vibrio cholerae or the Democratic Republic of Congo and why did you stop at 2017? Also, how did you screen all those results? If you type cholera and DRC into Google you get over 900,000 results.

Line 124 & 137: Why 2000-2017?

Is the epidemiological and surveillance data publicly available? Or did you have to request it from the Ministry of Health? If it is publicly available, a source may be helpful.

Line 263: You only looked at the weekly number of conflict events in endemic provinces, so it might be a bit of stretch to say that. I am not suggesting changing the analysis to include non-endemic conflict events but I think you need to include it as a limitation of the study. You briefly mention it on line 297, so maybe just expand that a little.

Line 294: Perhaps remove the term “world’s worst”, which is somewhat of an opinion.

In the whole discussion, WASH is never mentioned, which for a paper about cholera is a concern. For example, displacement doesn’t cause cholera, but instead people displaced without adequate clean water and sanitation. Additionally, conflict doesn’t cause cholera, but instead interruption in water supplies or damage to sanitation infrastructure. Trying to thread some of these themes into your discussion could really strengthen your narrative.

6. PLOS authors have the option to publish the peer review history of their article (what does this mean?). If published, this will include your full peer review and any attached files.

Reviewer #1: No

---

## [Author Response · Author response to Decision Letter 0]

9 Jan 2022

Line 58-62: “no particular attention has been paid to understanding the dynamics of the disease”, I am not sure this is justified. In term of cholera, we can definitely do more, but I am not quite sure you could say no attention has been paid to this. There are several excellent papers looking at cholera risk factors and transmission, some of which in the DRC and many you cite in the next paragraph.

Response: Thank you for this remark. This part was not clear in the former manuscript. The paragraph has been rewritten as follows: “Cholera was first reported in eastern DRC in 1978. Cases were determined to be imported from Kigoma in Tanzania to Kalemie, both bordering Lake Tanganyika, then the disease spread along the Great Lakes Region (GLR) (5,6). Over two decades later, the cholera control model in the DRC was essentially based on one-off response interventions in areas affected (5-7), and less attention has been paid to understanding the dynamics of the disease. Therefore, the role of certain factors, especially during inter-epidemic periods, in the recurrence of cholera outbreaks has not yet been investigated. Thus, the occurrence of epidemics seemed unpredictable, anarchic, even without spatio-temporal logic”. See page 3, lines 54-61.

Line 78: Would be good to include an extra sentence on why conflicts amplify the spread. E.g., disruption to WASH or healthcare, displacement etc.

Response: Thank you for your relevant suggestion. This part has been added in the manuscript as follows: “War and conflict lead to massive population displacement, the collapse of health systems, and the breakdown of surveillance, early warning and response systems (16). Conflict-affected populations are at risk of exposure to inadequate access to water, sanitation, and hygiene (WASH) facilities and resources (17,18), which further increases the risk of epidemics (19)”. See page 4, lines 71-81.

Line 93: Not sure about the ’s on health, double check that is grammatically sound.

Response: Thank you for this remark. The “ ‘s ” has been removed from “health” in the manuscript. See page 5, line 95.

The figures are great, informative and add to your narrative but they are a little blurry on my screen. Maybe just check the quality.

Response: Thank you for your suggestion. The quality of the figures has been improved using the Preflight Analysis and Conversion Engine (PACE) digital diagnostic tool.

Line 102-107: Why did you search only cholera and DRC, why not Vibrio cholerae or the Democratic Republic of Congo and why did you stop at 2017? Also, how did you screen all those results? If you type cholera and DRC into Google you get over 900,000 results.

Response: Thank you for these relevant observations. We used the full name of the country “Democratic Republic of Congo” and included “Vibrio cholerae” as keywords when searching for published articles or publicly available reports. This part has been reworded in the manuscript as follows: We used the following key words: “Cholera OR Vibrio cholerae” AND “Democratic Republic of the Congo”. See page 5 and lines 107-108.

We stopped at 2017 because the following years were respectively characterized by the continuity of the dynamics previously observed during the first part of 2018 and the relative stability of the situation across the country following the retraction of the epidemics in the endemic provinces of the Eastern Great Lakes region from the end of the same year.

Concerning the way we screened the results, we first included all the records found in PubMed, in the first 20 pages of Google scholar, and in the first 10 pages of Google. We then proceeded to the actual filtering. This part has been written in the 

manuscript as follows: “A list of 1,288 studies and reports were identified using the search terms. We removed 384 records due to duplication, and then we screened 904 records of which 889 were considered not relevant according to titles and abstracts or full texts that did not address the dynamics of the spread of cholera in and from eastern DRC. 15 studies and reports were eligible and detailed in S1 Table”. Please, see pages 8-9 and lines 181-185.

Line 124 & 137: Why 2000-2017? Is the epidemiological and surveillance data publicly available? Or did you have to request it from the Ministry of Health? If it is publicly available, a source may be helpful.

Response: Thank you for your questions. We considered the period 2000-2017 because the implementation of integrated surveillance of epidemic-prone diseases only started in 2000. Beyond 2017, there are continuity of the dynamics previously observed during the first part of 2018 and the relative stability of the situation across the country following the retraction of the epidemics in the endemic provinces of the Eastern Great Lakes region from the end of the same year.

This data is publicly available. The source has been given in the manuscript. See page 6, line 125.

Line 263: You only looked at the weekly number of conflict events in endemic provinces, so it might be a bit of stretch to say that. I am not suggesting changing the analysis to include non-endemic conflict events but I think you need to include it as a limitation of the study. You briefly mention it on line 297, so maybe just expand that a little.

Response: You are quite right to raise these observations. This part has been included as a limitation of the study and developed as follows: “Third, we just looked at the weekly number of conflict events in endemic provinces, without including non-endemic conflict events. Considering that endemic provinces account for 73% of reported conflict events in the country (27), two-thirds of which occur in North and South Kivu, we hypothesized that the exacerbation of conflicts in eastern DRC would be responsible for the spread of cholera to far-off areas in non-endemic provinces. Our results revealed that the most likely routes of spread involved in these dynamics would originate from endemic areas in North and South Kivu. However, some of the dynamics of spread outside out of endemic provinces were found to have been initiated from eastern areas less affected by conflicts. This further highlights the key role of population movements related to internally displaced persons on the one hand, and commercial activities on the other. There is need for futures studies to explore how conjunctural and structural population movements may affect the geographic spread of cholera from endemic eastern areas”. See pages 15-16 and lines 333-344.

Line 294: Perhaps remove the term “world’s worst”, which is somewhat of an opinion.

Response: Thank you for your suggestion. The term “world’s worst” has been removed in the manuscript. See page 14 and line 300.

In the whole discussion, WASH is never mentioned, which for a paper about cholera is a concern. For example, displacement doesn’t cause cholera, but instead people displaced without adequate clean water and sanitation. Additionally, conflict doesn’t cause cholera, but instead interruption in water supplies or damage to sanitation infrastructure. Trying to thread some of these themes into your discussion could really strengthen your narrative.

Response: Thank you for your relevant suggestions. They have been included in the manuscript as follows: “In our context, these dynamics of spread would be consecutive to the migration flows of thousands of internally displaced persons (IDPs) due to war and conflict situations. The latter also negatively impact the health system and other infrastructures that lose the capacity to provide even basic services (16). As a result, conflict-affected populations are at increased risk of food insecurity, shortages of potable water, and poor sanitation and hygiene practices in host sites due to rapid and massive relocation (17,18). It has been shown previously that a one-day interruption in water supply is followed by a substantial increase in the incidence rate of suspected cholera cases within 12 days (46)”. See page 14 and lines 302-309.

---

## [Editor Report · Decision Letter 1]

13 Jan 2022

Modalities and preferred routes of geographic spread of cholera from endemic areas in eastern Democratic Republic of the Congo

PONE-D-21-31003R1

Dear Dr. Kayembe,

We’re pleased to inform you that your manuscript has been judged scientifically suitable for publication and will be formally accepted for publication once it meets all outstanding technical requirements.

Kind regards,

Axel Cloeckaert

Academic Editor

PLOS ONE
---

## [Editor Report · Acceptance letter]

28 Jan 2022

PONE-D-21-31003R1 

Modalities and preferred routes of geographic spread of cholera from endemic areas in eastern Democratic Republic of the Congo 

Dear Dr. Kayembe:

I'm pleased to inform you that your manuscript has been deemed suitable for publication in PLOS ONE. Congratulations! Your manuscript is now with our production department. 

Kind regards, 

on behalf of

Dr. Axel Cloeckaert 

Academic Editor

PLOS ONE